# Case Series of Rare Fungal Keratitides: Experiences from a Quaternary Eye Hospital in Sydney, Australia

**DOI:** 10.3390/jof9050589

**Published:** 2023-05-18

**Authors:** Rachel Xuan, Sheng Chiong Hong, Tanya Trinh, Minas T. Coroneo, Constantinos Petsoglou

**Affiliations:** 1Department of Ophthalmology, Sydney and Sydney Eye Hospital, Sydney, NSW 2000, Australia; rachel.xuan29@gmail.com (R.X.);; 2Save Sight Institute, University of Sydney, Sydney, NSW 2000, Australia; 3Department of Ophthalmology, Prince of Wales Hospital, Randwick, Sydney, NSW 2031, Australia; 4School of Medicine, University of New South Wales, Kensington, Sydney, NSW 2033, Australia

**Keywords:** fungal keratitis, immunocompetent, caspofungin, posaconazole, antifungal, debridement, evisceration, fungal endophthalmitis

## Abstract

The present article reports on the management of six different and rare cases of fungal keratitides, two of which have never been documented in previous literature. This is a case series of six patients with rare fungal keratitides managed at a quaternary eye referral unit, Sydney Eye Hospital, Australia over a period of 7 months (May to December, 2022). The order of occurrence of fungi isolated was *Scedosporium apiospermum*, *Lomenstospora prolificans*, *Cladosporium* spp., *Paecilomyces*, *Syncephalastrum racemosum* and *Quambalaria* spp. A combination of medical and surgical interventions was employed, including topical and systemic anti-fungal therapy, with one requiring therapeutic penetrating keratoplasty and another eventuating in evisceration. Two patients were successfully treated with corneal debridement and two others required pars plana vitrectomy with anterior chamber washout. It is important to remain vigilant with monitoring patient symptoms and correlating with clinical signs to guide antifungal therapy even in the context of confirmed culture and sensitivity results.

## 1. Introduction

Over the past decades, the prevalence of ocular fungal infections has increased substantially [1]. Risk factors for fungal keratitis have been well studied in the literature, including increasing use of steroids, prolonged contact lens wear, broad-spectrum antibiotics, trauma with vegetative matter, ocular surface disease, corneal surgery and systemic immunosuppression [2,3,4,5]. Despite the heightened awareness among clinicians and advances in laboratory diagnostics, the treatment of fungal disease remains a challenge [6]. This is in part due to an expanding list of fungal pathogens and relatively limited available therapeutic agents [7]. A surgical approach is often required when medical therapy is inadequate [8].

The challenges of accurate and timely clinical diagnosis are further complicated in cases of partial treatments, mixed infections and lack of routine sensitivity testing of fungal isolates in the laboratory [7,9]. More than 70 species of fungi have been reported as pathogenic to the human cornea, with the course of therapy often prolonged, responding poorly to standard medical therapy and resulting in poor clinical outcomes [6]. This case series presents rare and emerging, culture-proven cases of fungal keratitides, of which two have not previously been documented in the literature. By outlining the clinical characteristics, laboratory investigations, treatment and outcomes of these cases, we aim to increase awareness and to improve the management of recalcitrant, rare fungal keratitis.

## 2. Methods

This retrospective case series from May 2022 to November 2022 was conducted at Sydney Eye Hospital, a quaternary referral unit for eye diseases located in Sydney, Australia. Patients were included if their clinical presentation was consistent with fungal keratitis and they had a positive laboratory culture for fungi. Demographic data, ocular risk factors, clinical presentation and management were extracted from medical record notes. All of the patients included in this paper have provided written consent.

All patients had initially undergone a corneal scraping, either prior to referral to Sydney Eye Hospital or on direct presentation. Corneal sampling at Sydney Eye Hospital was carried out using aseptic technique with sterile surgical blades, sterile spatulas or disposable needles. Samples were immediately placed onto two glass slides for microscopy and then inoculated on two blood agars, chocolate agar, Sabouraud’s agar slope containing chloramphenicol and gentamicin and cooked meat medium. Corneal swabs were also taken for PCR for the detection of herpes simplex virus, varicella zoster virus and acanthamoeba DNA. In vivo confocal microscopy was occasionally used to assist in detecting recurrence of fungal growth. Where relevant, an anterior chamber (AC) tap was performed with a 25 G needle on a 1 mL syringe via sterile technique in the operating theatre. Corneal debridement was carried out in an outpatient setting with the patient positioned at a slit lamp after topical oxybuprocaine instillation.

Best-corrected visual acuity (BCVA) was recorded at presentation to the hospital and upon discharge or at the last follow up. BCVA was measured using a Snellen projector chart in the hospital, unaided or aided with their usual means of correction.

## 3. Results

Table 1 summarises the demographic data of patients, their presenting problem, risk factors, past ocular history, comorbidities, fungal organism isolated, treatment and outcome. In total, 50% of the patients had concomitant bacterial keratitis.

Table 2 shows the culture and sensitivity testing.

### 3.1. Case 1

A 66-year-old cattle farmer was referred for right eye *Pseudomonas* keratitis. He was also on immunosuppressant medications including mycophenolate 1000 mg twice daily and cyclosporin 50 mg twice daily for graft-versus-host-disease (GvHD) following a previous allogenic stem cell transplant for non-Hodgkin’s lymphoma. Slit lamp examination revealed a 5.5 by 5.5 mm infiltrate with complete corneal fluorescein staining, anterior chamber inflammation and a 0.8 mm hypopyon. His corneal scrape grew *Syncephalastrum racemosum* (sensitive to amphotericin B, itraconazole and posaconazole) and *Bacillus* sp., which was susceptible to chloramphenicol. Corneal swabs also detected varicella zoster virus deoxyribonucleic acid (DNA) treated with systemic valacyclovir. Seven months after treatment, the patient had a residual central corneal plaque and scarring with neovascularization (Figure 1) on autologous serum eye drops. He remained systemically well; hence, no fungal blood cultures or systemic fungal markers were performed.

### 3.2. Case 2

A 59-year-old contact lens wearer presented with a red and painful left eye after a day of gardening. Her BCVA was 6/6 but deteriorated to 6/60. Slit lamp examination revealed a 3.5 by 2.5 mm infiltrate with overlying epithelial defect, microcystic oedema and Descemet membrane folds. A left eye exam was unremarkable. Whilst on topical antibiotic drops, she developed multiple further infiltrates with a feathered border as well as multiple circumferential, satellite-like, subepithelial, cystic lesions with surrounding stromal ring and trace cells in the anterior chamber. Her corneal scrape returned positive results of *Scedosporium apiospermum* (sensitive to voriconazole minimal inhibitory concentration, MIC, level 0.5) and *Paenibacillus* sp. (growth in liquid enrichment medium only). After 13 weeks of treatment, a 1.5 mm by 1.5 mm white infiltrate was noted at follow up and confocal microscopy was used to confirm fungal growth (Figure 2). She then started topical caspofungin 0.5%. and the suspected area of fungal recurrence was also therapeutically debrided three times over a course of 6 weeks. At week four after the last debridement, the infiltrate completely resolved (Figure 3).

### 3.3. Case 3

A 24-year-old male was referred with a left eye hypopyon, anterior chamber inflammation and vitritis on B scan ultrasonography. Initially, fungus was isolated from the corneal glue used to patch his atraumatic corneal perforation 2 months prior (Figure 4A). He received a corneal graft 2 weeks later and voriconazole was ceased two weeks post grafting. In the week after ceasing voriconazole, he developed a white cataract with graft neovascularisation and fungal invasion of the graft (Figure 4B). Despite an intraoperative washout with antibiotics, the patient experienced worsening eye pain and re-developed a hypopyon and corneal graft oedema. He was then transferred to Sydney Eye Hospital where he was diagnosed with endophthalmitis and received a left eye lensectomy, pars plana vitrectomy, intravitreal injection of voriconazole and anterior chamber washout. His vitreous fluid culture grew *Purpureocillium lilacinum* for which he was commenced on topical voriconazole (MIC level 0.25) and oral posaconazole (MIC level 0.5). Three months after presentation, his vision was hand movement and clinical examination and demonstrated ongoing corneal oedema with anterior chamber inflammation but no hypopyon.

### 3.4. Case 4

A 45-year-old female presented with left eye keratitis in the context of recent eye irritation, thought to be due to a herpetic infection three months prior. She had laser-assisted in situ keratomileusis (LASIK) 12 years ago in both eyes but no other ocular or medical conditions. She failed topical and oral voriconazole treatment and hence was commenced on topical caspofungin 5% and amphotericin B 0.15%. Her corneal scrape demonstrated growth of *Cladosporium* spp. However, after one week of dual antifungal therapy, there was a persisting infiltrate at the edge of the LASIK interface and she underwent two therapeutic corneal debridements (3 and 4 weeks following presentation) to achieve a visual acuity (VA) of 6/6 after 4 weeks of treatment. Repeat corneal scrapes from the debridement were negative for any fungal growth.

### 3.5. Case 5

A 55-year-old female contact lens wearer presented to Sydney Eye Hospital with a red, painful and photophobic right eye. Five days prior to presentation she had an organic foreign body removed by the local optometrist. Her clinical examination revealed a 2 mm × 1.1 mm infiltrate with a long inferior spoke of 1.4 mm × 0.5 mm with an overlying epithelial defect, anterior chamber inflammation but no hypopyon. Her corneal scrape grew *Lomentospora prolificans* and *Micrococcus luteus* (sensitive to chloramphenicol). Ten days after presentation, her BCVA was 6/9 with an epithelial scar (Figure 5B) and resolved corneal ulcer.

### 3.6. Case 6

An 81-year-old female was referred with a right corneal ulcer, hypopyon and anterior chamber inflammation, one month following a trabeculectomy surgery. Her vitreous fluid culture was positive for *Quambalaria* sp. and *Penicillium* sp. Despite empirical therapy and intraoperative washout, by day 6 of treatment, the infiltrate had progressed to 80% of the entire cornea with signs of worsening endophthalmitis and the decision was made for evisceration with placement of an orbital implant.

## 4. Discussion

The geography of Sydney falls in the temperate climate zone with warm to hot summers and varying rainfall that peaks in the first few months of year. In recent years, Sydney has been subjected to both heavy rainfall and hot weather which may be a contributing factor to increased cases of fungal keratitides [10]. In this environment, those who work with agriculture or have farming occupations are at an increased risk of exposure to fungi [4,11]. All patients in this case series had experienced ocular trauma with two having had contact with an organic foreign body and one being a cattle farmer in occupation.

Another consideration is that all of these cases occurred during the COVID-19 pandemic. It is well known that immune-suppression either directly related to COVID-19 infection or the treatment of COVID may predispose to infection. The incidence or characteristics of microbial keratitis did not change in some centres [12]; however, cases of severe rapidly progressive fungal keratitides and endophthalmitis post-COVID-19 infection have been reported in individual cases [13]. Patient 4 developed COVID-19 six months before presenting with *Cladosporium* keratitis; however, she did not require any treatment for her COVID-19 infection. Patient 1 developed COVID-19 four months after presenting with *Syncephalastrum* keratitis.

This case series provides real-world perspective outlining the treatment and clinical progression of patients with rare and previously not documented fungal keratitis. To the best of the authors’ knowledge, this is the first report of *Quambalaria* sp. and *Syncephalastrum racemosum* keratitis.

*Syncephalastrum* is a rare, opportunistic fungal species with few case reports of cutaneous and respiratory infections in immunocompromised patients [14,15,16]. It is likely that a host of underlying systemic risk factors including a significant ocular history and surface disease, recent topical steroid use, demographic background and non-compliance with treatment predisposed case 5 to developing *Syncephalastrum racemosum*. There is no current documented keratitis of this causative agent. Similarly, there are limited data from reported cases suggesting that *Quambalaria* is a pathogen primarily impacting systemically unwell individuals [17,18,19]. The genus *Quambalaria* is a hyaline basidiomycete predominantly known as non-pathogenic to plants including *Corymbia* and *Eucalyptus* species in Australia [20,21]. We report the first case of *Quambalaria* sp. isolated from the cornea of an immunocompetent patient. Both species were associated with poor clinical outcomes. We described these cases to emphasise the importance of considering rare fungal causes where the infection runs an unusual course or is particularly aggressive. It is also essential to maintain a high degree of clinical suspicion even in immunocompetent individuals.

To assist in the diagnosis of fungal keratitis, the use of in vivo confocal microscopy (IVCM) has been recommended as a rapid and non-invasive method [22]. Recent studies have further demonstrated its utility in guiding anti-fungal therapy or signalling when a change in therapy is required [23,24]. Similarly in case 2, IVCM was used to confirm the recurrence of *Scedosporium* growth, and the topical therapy was changed from voriconazole to caspofungin. Additionally, IVCM could also aid in the decision for surgical intervention when there is progression of disease despite medical treatment [25].

This case series also documents the first successful treatment of *Cladosporium* keratitis with caspofungin. Caspofungin is an echinocandin which acts by inhibiting the synthesis of glucan in the fungal cell wall causing osmotic imbalance and cell lysis [26]. There are limited data on the use of echinocandins to treat fungal keratitis in humans with few reports of topical caspofungin used in *Alternaria* [27], *Fusarium* [28] and *Candida albicans* [29] keratitis. These cases suggest the potential usefulness of topical caspofungin in treating fungal keratitis; however, more evidence on its clinical utility is needed before it can be incorporated into standard treatment regimens [30]. At the Sydney Eye Hospital, caspofungin 0.5% eye drops are compounded by the pharmacy by adding one caspofungin 50 mg vial to 10 mL of sodium chloride 0.9% solution, resulting in four bottles of 2.5 mL caspofungin 0.5% eye drops. These have a 14-day shelf-life and last only 3.5 days once opened. The drug is reconstituted in 0.9% sodium chloride instead of sterile water due to the low osmolality of the final product when prepared with water; hence, this formula replaces sterile water with 0.9% sodium chloride [31].

Previous studies have reported successful treatments of both fungal species, even in recurrent cases, with voriconazole [32,33,34]. Interestingly, both patients in our series failed voriconazole treatment despite a positive culture result demonstrating *Scedosporium apiospermum* and *Purpureocillium lilacinum* sensitivity to voriconazole (MIC level 0.5 μg/mL). This could be attributed to an increasing fungal resistance to voriconazole with a recent study (2022) demonstrating a 1.02-fold increase in resistance per year [35]. Hence, it is important to remain vigilant with monitoring patient symptoms and correlating with clinical signs to guide antifungal therapy even in the context of confirmed culture and sensitivity results. In cases not responding to conventional treatments, alternative therapies should therefore be considered.

Posaconazole is a newer triazole antifungal that is usually well tolerated and has the added benefit of extended side chains that provide additional points of contact with the azole target, CYP51, compared to voriconazole [36,37]. There have been four other reports of refractive *Purpureocillium lilacinus* treated successfully with posaconazole [38,39,40,41]. Additional case reports further highlight the clinical efficacy of posaconazole in cases of recalcitrant fungal keratitides not responding to conventional therapies [40,42].

This is also the first case of *Lomentospora prolificans* (formerly *Scedosporium prolificans*) [43] keratitis successfully treated with solely medical management using topical natamycin.

There have been few published reports of *Scedosporium prolificans* sclerokeratitis and keratouveitis treated with topical antifungals; however, all required surgical intervention [44,45,46,47,48]. One case of *Scedosporium prolificans* keratouveitis was only treated with ciprofloxacin for a concomitant bacterial keratitis and the patient recovered prior to the release of the fungal culture results [44]. The management of rare fungal keratitides is further complicated by the continuous changes to fungal nomenclature as molecular technologies replace conventional phenotypic methods of classification [49]. Hence, treatment regimens for fungal keratitis, particularly recalcitrant cases, should involve discussion with infectious diseases physicians.

Adjunctive topical steroids are often used in microbial keratitis to suppress the inflammatory response and reduce the sequalae of corneal scarring and neovascularisation [50,51]. However, its use can potentiate and exacerbate the progression of fungal infections; hence, it must not be commenced until a causative agent is found and/or the patient has shown a favourable response to antibiotic/antifungal therapy [6,51]. An alternative treatment to control inflammation associated with fungal keratitis is topical cyclosporin, a calcineurin inhibitor and antifungal metabolite with antifungal properties [52]. It has been validated in the use of increasing corneal graft survival post penetrating keratoplasty for fungal keratitis as well as being synergistically combined with azole agents in treating resistant fungi [52,53]. Unlike corticosteroids, which cause widespread suppression of the host’s immune response, cyclosporin primarily blocks T-cell responses but has minimal to no effect on macrophages at the same concentration [54]. In the case of *Purpureocillium lilacinum* keratitis which progressed to endophthalmitis, cyclosporin was used instead of steroids for its selective inhibition of the immune system. This case series further highlights the multifaceted role of cyclosporin in fungal keratitis in the context of a patient with a corneal graft, risk of rejection and the need to reduce inflammation.

Topical antifungal therapy in our series reflects that of previous studies conducted in Australia, with topical natamycin used as standard empirical therapy for filamentous fungal keratitis [4,5]. It is also preferred due to its commercial availability and ease of access. If there was progressive keratitis, a second antifungal agent was added. However, two thirds of our case series showed little response even to dual, topical and/or systemic antifungal therapy, further requiring antifungal injections or surgical intervention. In keeping with previous studies, when medical therapy failed in our cases, surgical interventions such as corneal debridement, therapeutic keratoplasty, anterior chamber washout and evisceration were then performed [5,55,56]. Corneal debridement has been demonstrated to provide a faster clinical resolution due to its facilitation of topical medical penetration with no reduction in visual potential and more accurate acquisition of pathogens than conventional scraping [57]. It is an effective and safe way to treat fungal keratitis, especially in recalcitrant cases unresponsive to medical therapy [58,59,60].

In summary, our case series describes the use of medical and surgical intervention in managing rare and recalcitrant fungal keratitis. The series will assist with local treatment and management, especially for fungal organisms not previously documented.

## 5. Conclusions

The treatment and management of recalcitrant fungal keratitis can be protracted, especially in the context of limited standard protocols and delayed diagnosis. Hence, it is necessary that clinicians closely monitor and correlate symptoms with clinical progression in guiding antifungal therapy. Furthermore, in fungal keratitis which is progressive or non-responsive to medical therapy, alternate modalities such as corneal debridement, penetrating keratoplasty and evisceration should be considered.

## Figures and Tables

**Figure 1 jof-09-00589-f001:**
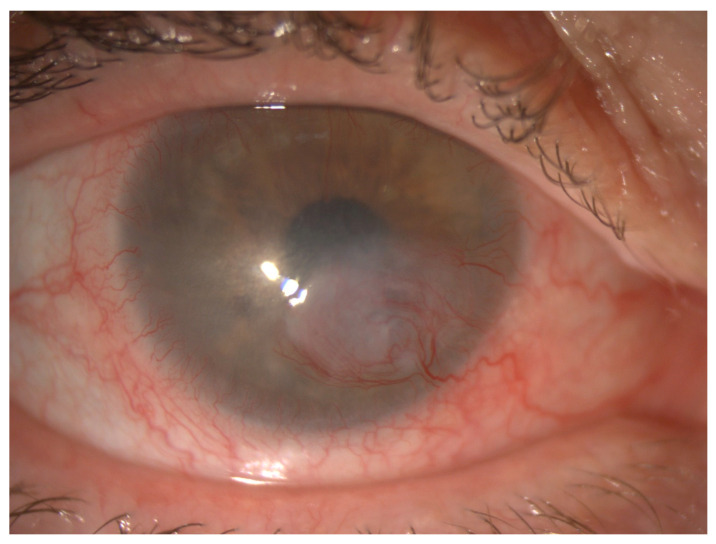
Central corneal plaque with neovascularisation and stem-cell failure in the right eye.

**Figure 2 jof-09-00589-f002:**
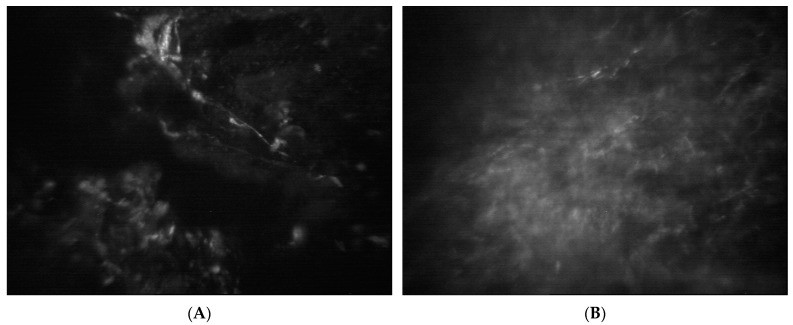
Confocal microscopy images of *Scedosporium* keratitis. (**A**,**B**) Representative images showing hyper-reflective linear structures with acute angle branching, typical of fungal hyphae. Images are provided at a scale of 360 × 430 micro-millimetre.

**Figure 3 jof-09-00589-f003:**
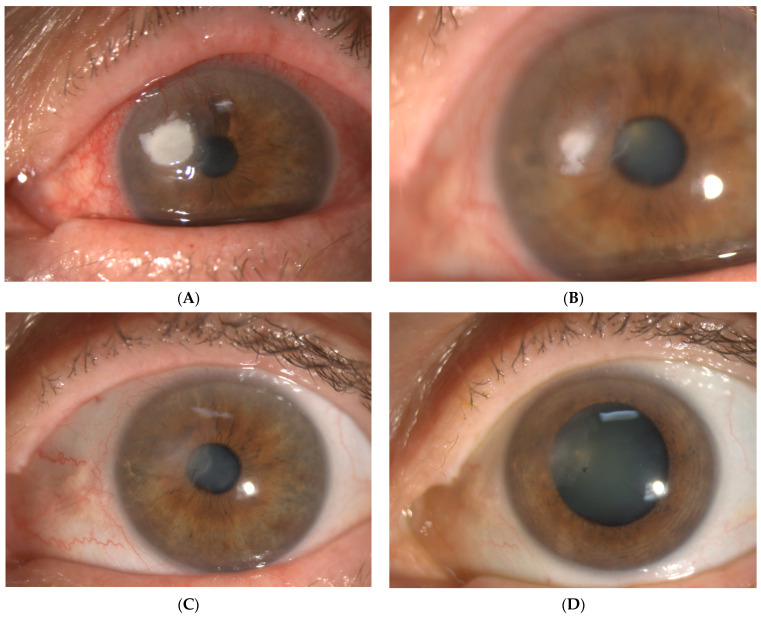
(**A**) Left eye infiltrate prior to debridement. (**B**) Left eye infiltrate one month after first debridement. (**C**) Left eye infiltrate one month after second debridement. (**D**) Left eye infiltrate one month after third debridement.

**Figure 4 jof-09-00589-f004:**
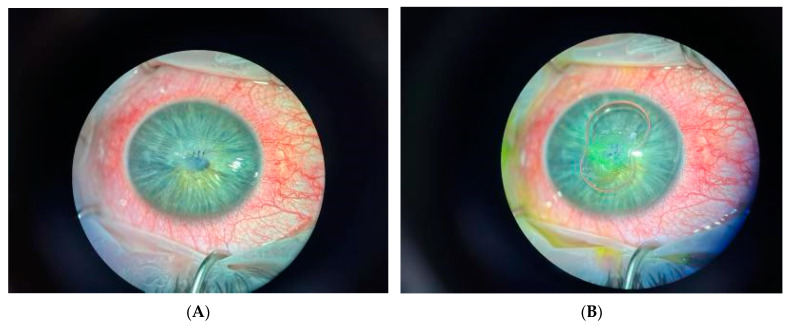
(**A**) Intraoperative image of corneal suture for spontaneous left eye perforation from keratoconus. (**B**) Subsequent development of left eye fungal keratitis.

**Figure 5 jof-09-00589-f005:**
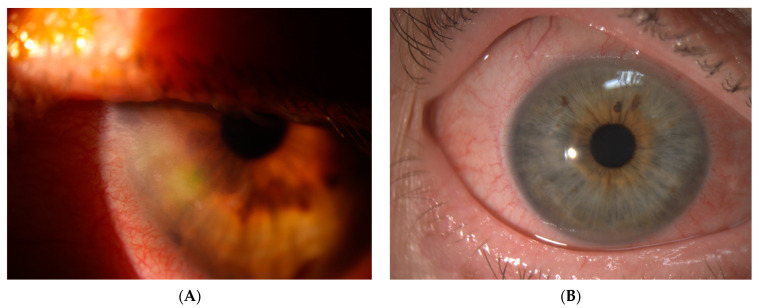
(**A**) Left *Cladosporium* keratitis at edge of LASIK interface. (**B**) Residual corneal scar in right eye from *L. prolificans* keratitis.

**Table 1 jof-09-00589-t001:** Cases.

Case	Age	Sex	Eye	Immunosuppression or Risk Factor	Fungal Organism (Specimen)	Past Ocular History	Comorbidities	Treatment		BVCA
								Medical	Surgical	
**1**	66	M	RE	Mycophenolate, cyclosporin. Cattle farmer. Steroids for ocular herpes zoster (right eye). Medication non-compliance	*Syncephalastrum racemosum* *Bacillus* sp. (corneal scrape)	Right eye pseudophakic, Glaucoma Shingles	Non-Hodgkin’s lymphoma with allogenic stem cell transplant	Topical voriconazole + ciclosporin + amphotericin B	Nil	6/120-1
**2**	59	F	LE	Contact lens wear. Gardening. Steroids for ocular rosacea. Gestational diabetes	*Scedosporium apiospermum* *Paenibacillus* sp. (corneal scrape)	Nil	Gestational diabetes (recent HbA1c 6.1%)	Topical caspofungin.	Therapeutic corneal debridement (outpatient)	6/9+1
**3**	24	M	LE	Significant ocular history	*Purpureocillium lilacinum* (vitreous fluid)	Keratoconus Recurrent keratitis Atraumatic corneal perforation	Nil	Topical voriconazole + PO posaconazole	Total penetrating keratoplasty PPV/IVI/AC washout + lensectomy	6/60
**4**	45	F	LE	Nil	*Cladosporium* spp. (corneal scrape)	LASIK	Nil	Topical amphotericin B + topical caspofungin.	Therapeutic corneal debridement (outpatient)	6/6
**5**	55	F	RE	Contact lens wearer	*Lomentospora prolificans Micrococcus luteus* (corneal scrape)	Nil	Nil	Topical natamycin	Nil	6/9-2
**6**	81	F	RE	Right eye trabeculectomy one month prior	*Quambalaria* sp. *Penicillium* sp. (corneal scrape)	Glaucoma	Hypertension hypothyroidism	PO moxifloxacin + Topical ofloxacin + gentamicin + ceftazidime	PPV/IVI/AC washout Evisceration	Eviscerated

Abbreviations: PO oral; IC intracameral; SC subconjunctival; BCVA best corrected visual acuity; TPK therapeutic penetrating keratoplasty; LASIK laser assisted in situ keratomileusis; PPV pars plana vitrectomy; AC anterior chamber; HTN hypertension; IVI intravitreal injection.

**Table 2 jof-09-00589-t002:** MIC values for fungal cultures in μg/mL.

	Case 1	Case 2	Case 3	Case 4	Case 5	Case 6
	*Syncephalastrum racemosum*	*Scedosporium apiospermum*	*Purpureocillium lilacinum*	*Cladosporium* spp.	*Lomentospora prolificans*	*Quambalaria* sp.
**Amphotericin B**	0.5	>8	>8	N/A	>8	N/A
**Fluconazole**	>256	8	16		>256	
**Flucytosine**	>64	>64	>64		>64	
**Isavuconazole**	4	2	1		>8	
**Itraconazole**	0.5	1	>16		>16	
**Micafungin**	>8	0.25	0.12		>8	
**Posaconazole**	0.5	1	0.5		>8	
**Voriconazole**	>8	0.5	0.25		>8	

The Clinical and Laboratory Standards Institute Guidelines were used for the fungal cultures performed in the case series.

## Data Availability

Data is unavailable due to privacy or ethical restrictions.

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
