# Peer review of "Case Series of Rare Fungal Keratitides: Experiences from a Quaternary Eye Hospital in Sydney, Australia"

_jof, 2023, doi:10.3390/jof9050589_

Round 1
Reviewer 1 Report
In this single center case series from Australia, the study authors report on fungal keratitides occurring due to different species in six distinct patients. They expand on their case histories and highlight unique points about some of the more uncommon fungi. Overall, the case report is interesting and would add to the literature around these fungal infections. However, certain important points need to be clarified in the case histories.
Major Comments
1. The findings of fungal keratitis and ocular symptoms (particularly in immunosuppressed individuals) would warrant evaluation for systemic fungal infections, testing of fungal markers and imaging studies. Is there available information on these points for all the patients, or atleast for patients 1 and 2? It would be educative to include the fungal blood culture results, serum beta-D-Glucan and serum Aspergillus antigen results and MRI/CT imaging studies if available. A simple sentence saying “Fungal blood cultures were negative” would also suffice, but I think it would be worthwhile to include these points for the patients.
2. It would be helpful to include the dosages of the immunosuppressant medications (if available) for the patients 1 and 2, as they appear to be the most immunosuppressed from the six patients. This would also be beneficial to non-ophthalmologic medical professionals taking care of these patients.
3. Given the underlying immunosuppressed status of patient 1, did he also receive treatment for Bacillus and the VZV DNA noted in the corneal scraping or were they thought to be colonizers? It would be educative to explain that
4. Adding to the above point, it would be helpful to include information related to the underlying gestational diabetes mellitus (DM) history of patient 2 and her most recent hemoglobin A1c level. Both the use of corticosteroids for ocular rosacea and the underlying DM likely contributed to the fungal infection, but in the current table this information appears to be incomplete.
5. The study authors note in the Discussion that their cases occurred in the COVID-19 pandemic and one patient (I presume patient 4) developed Cladosporium keratitis after COVID-19. The study authors should highlight her clinical course and therapies she received for COVID-19.
6. In Table 3, the study authors have documented MIC values of different antifungal agents against their isolates and have also documented how two patients had failed voriconazole treatment despite the reported susceptibility. It would be helpful to indicate the guidelines (CLSI or EUCAST) guidelines on which the susceptibility was determined and also to include that information (as additional columns, if feasible) in Table 3.
Minor Comments
1. Page 3 of 12, Table 2 – please provide expansions of the multiple abbreviations used in the Table, such as PPV, AC, IVI, TPK. I understand that LASIK and HTN may be standard, but they should also include these if possible.
2. Page 4 of 12, Case 1 – please specify whether the patient had undergone autologous or allogeneic hematopoietic stem cell transplant. The underlying extent of immunosuppression varies between these types of transplants.
Author Response
Reply is attached in this word document

Reviewer 2 Report
This retrospective study reported the rare fungal keratitides at a quaternary eye referral unit, Sydney Eye Hospital, Australia. The study is interesting; however, I have several suggestions.
1. Please rewrite the result section and try to summarize the reported case in clinical and microbiological parts rather detailed describe each case, which was showed in table 2.
2. Please move the information of table 1 into table 2 and then delete table 1.
3. Add the concomitant bacterial information in the table 2.
4. Check all the abbreviations in the text and table. Please make sure their full name should be spell out for the first presence.
Author Response
see attached reply

Round 2
Reviewer 1 Report
Overall, the study authors have incorporated the suggestions and addressed the comments as previously raised.
One minor comment that I would like to raise is that apart from stating that fungal blood cultures were not drawn, it would also be educative to indicate that serum fungal markers were not tested.
Author Response
We have amended to include the comment about fungal markers.